# Fine-Grained Dynamic Head for Object Detection

Lin Song[1]    Yanwei Li[2]    Zhengkai Jiang[3]    Zeming Li[4]
Hongbin Sun[1]*    Jian Sun[4]    Nanning Zheng[1]
[1] College of Artificial Intelligence, Xi'an Jiaotong University
[2] The Chinese University of Hong Kong
[3] Institute of Automation, Chinese Academy of Sciences
[4] Megvii Inc. (Face++)
stevengrove@stu.xjtu.edu.cn, ywli@cse.cuhk.edu.hk, jiangzhengkai2017@ia.ac.cn,
{hsun, nnzheng}@mail.xjtu.edu.cn, {lizeming, sunjian}@megvii.com

## Abstract

The Feature Pyramid Network (FPN) presents a remarkable approach to alleviate the scale variance in object representation by performing instance-level assignments. Nevertheless, this strategy ignores the distinct characteristics of different sub-regions in an instance. To this end, we propose a fine-grained dynamic head to conditionally select a pixel-level combination of FPN features from different scales for each instance, which further releases the ability of multi-scale feature representation. Moreover, we design a spatial gate with the new activation function to reduce computational complexity dramatically through spatially sparse convolutions. Extensive experiments demonstrate the effectiveness and efficiency of the proposed method on several state-of-the-art detection benchmarks. Code is available at https://github.com/StevenGrove/DynamicHead.

## 1   Introduction

Locating and recognizing objects is a fundamental challenge in the computer vision domain. One of the difficulties comes from the scale variance among objects. In recent years, tremendous progress has been achieved on designing architectures to alleviate the scale variance in object representation. A remarkable approach is the pyramid feature representation, which is commonly adopted by several state-of-the-art object detectors [1–5].

Feature Pyramid Network (FPN) [6] is one of the most classic architectures to establish pyramid network for object representation. It assigns instances to different pyramid levels according to the object sizes. The allocated objects are then handled and represented in separate pyramid levels with corresponding feature resolutions. Recently, numerous methods have been developed for better pyramid representations, including human-designed architectures (*e.g.,* PANet [7], FPG [8]) and searched connection patterns (*e.g.,* NAS-FPN [9], Auto-FPN [10]). The above-mentioned works lay more emphasis on instance-level *coarse-grained* assignments and considering each instance as a whole indivisible region. However, this strategy ignores the distinct characteristic of the *fine-grained* sub-regions in an instance, which is demonstrated to improve the semantic representation of objects [11–18]. Moreover, the conventional head [2, 3, 19–22] for FPN encodes each instance in a single resolution stage only, which could ignore the small but representative regions, *e.g.,* the hands of the person in Fig. 1(a).

In this paper, we propose a conceptually novel method for fine-grained object representation, called *fine-grained dynamic head*. Different from the conventional head for FPN, whose prediction only associates with the single-scale FPN feature, the proposed method *dynamically* allocates *pixel-level*

---

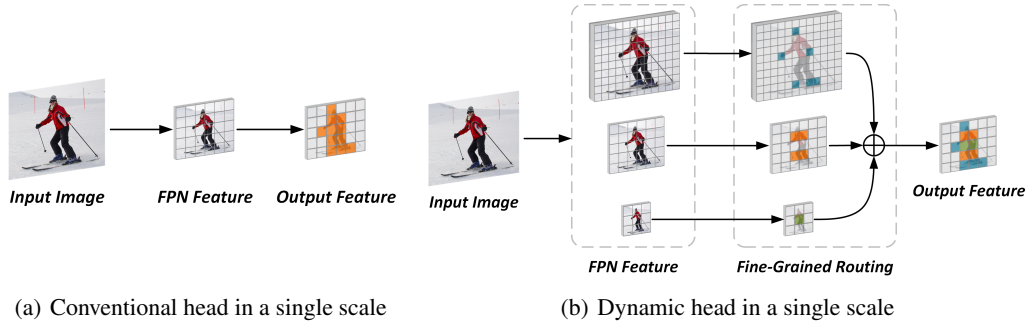

(a) Conventional head in a single scale      (b) Dynamic head in a single scale

Figure 1: Comparisons between the conventional head and the proposed dynamic head in a single scale. In the conventional head, the output feature of an instance only associates with a single-scale FPN feature. With the fine-grained routing process, the proposed dynamic head selects a combination of *pixel-level* sub-regions from multiple FPN stages.

sub-regions to different resolution stages and aggregates them for the finer representation, as presented in Fig. 1(b). To be more specific, motivated by the coarse-grained dynamic routing [23–26], we design a fine-grained dynamic routing space for the head, which consists of several fine-grained dynamic routers and three kinds of paths for each router. The fine-grained dynamic router is proposed to conditionally select the appropriate sub-regions for each path by using the data-dependent spatial gates. Meanwhile, these paths are made up of a set of pre-defined networks in different FPN scales and depths, which transform the features in the selected sub-regions. Different from the coarse-grained dynamic routing methods, our routing process is performed in pixel-level and thus achieves fine-grained object representation. Moreover, we propose a new activation function for the spatial gate to reduce computational complexity dramatically through spatially sparse convolutions [27]. Specifically, with the given resource budgets, more resources could be allocated to "hard" positive samples than "easy" negative samples.

Overall, the proposed dynamic head is fundamentally different from existing methods for head designs. Our approach exploits a new dimension: *dynamic routing mechanism is utilized for fine-grained object representation with efficiency*. The designed method can be easily instantiated on several FPN-based object detectors [2, 3, 5, 20, 21] for better performance. Moreover, extensive ablation studies have been conducted to elaborate on its superiority in both effectiveness and efficiency, which achieve consistent improvements with little computational overhead. For instance, with the proposed dynamic head, the FCOS [3] based on the ResNet-50 [28] backbone attains 2.3% mAP absolute gains with less computational cost on the COCO [29] dataset.

## 2 Method

In this section, we first propose a routing space for the fine-grained dynamic head. And then, the fine-grained dynamic routing process is elaborated. At last, we present the optimization process with resource budget constraints.

### 2.1 Fine-Grained Dynamic Routing Space

FPN-based detection networks [2, 3, 20, 21] first extract multiple features from different levels through a backbone feature extractor. And then, semantically strong features in low-resolution and semantically weak features in high-resolution are combined through the top-bottom pathway and lateral connections. After obtaining features from the pyramid network, a head is adopted for each FPN scale, which consists of several convolution blocks and shares parameters across different scales. The head further refines the semantic and localization features for classification and box regression.

To release pixel-level feature representation ability, we propose a new fine-grained dynamic head to replace the original one, which is shown in Fig. 2. Specifically, for $n$-th stage, a space with $D$ depths and three adjacent scales is designed for a fine-grained dynamic head, namely *fine-grained dynamic routing space*. In this routing space, the scaling factor between adjacent stages is restricted to 2. The

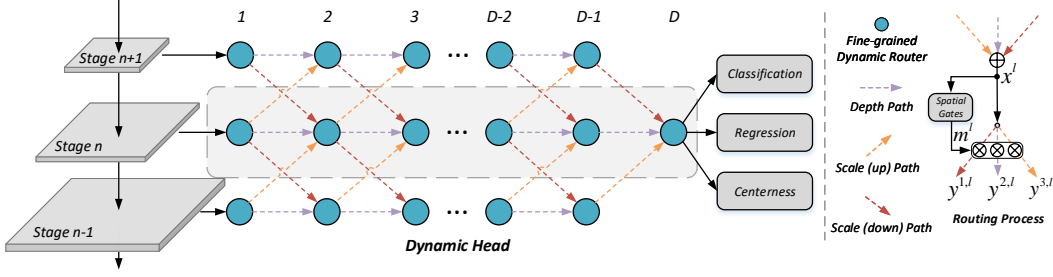

Figure 2: The diagram of the routing space for the fine-grained dynamic head. 'Stage n-1', 'Stage n' and 'Stage n+1' represent three adjacent FPN scales, respectively. The dynamic head (marked by the dashed rectangle) for each stage includes $D$ fine-grained dynamic routers, where each router has up to three alternative paths. The router first aggregates the multiple input features by performing element-wise accumulation. And then, for each *pixel-level* location, the router dynamically selects subsequent paths, which is elaborated in 'Routing Process'.

basic unit, called *fine-grained dynamic router*, is utilized to select subsequent paths for each pixel conditionally. In order to take advantages of distinct properties of the different scale features, we propose three kinds of paths with different scales for each router, which are elaborated in Sec. 2.2.3. Following common protocols [2, 3, 20, 21], we adopt the ResNet [28] as the backbone. The feature pyramid network is configured like the original design [6] with simple up-sampling operations, which outputs augmented multi-scale features, *i.e.,* {P3, P4, P5, P6, P7}. The channel number of each feature is fixed to 256.

## 2.2 Fine-Grained Dynamic Routing Process

Given the routing space with several individual nodes, we propose *fine-grained dynamic routers* for each router to aggregate multi-scale features by performing element-wise accumulation and choose pixel-level routing paths. This routing process of a dynamic router is briefly illustrated in Fig. 2.

### 2.2.1 Fine-Grained Dynamic Router

Given a node $l$, the accumulation of multiple input features is denoted as $\mathbf{x}^l = \{\mathbf{x}^l_i\}^N_{i=1}$ with $N(N = H \times W)$ pixel-level locations and $C$ channels. We define a set of paths according to the adjacent FPN scales, which is denoted as $\mathbf{F} = \{f^l_k(\cdot)|k \in \{1, ..., K\}\}$. To dynamically control the on-off state of each path, we set up a *fine-grained dynamic router* with $K$ *spatial gates*, where the output of the spatial gate is defined as *gating factor*. The gating factor of the $k$-th spatial gate in $i$-th ($i \in \{1, ..., N\}$) location can be represented by

$$m^{k,l}_i = g^l_k(\mathbf{x}^l_i; \theta^l_k),\tag{1}$$

where $\theta^l_k$ denotes the parameters of the partial network corresponding to the $k$-th spatial gate and shares parameters in each location. The conventional evolution or reinforcement learning based methods [4, 9, 10] adopt metaheuristic optimization algorithm or policy gradient to update the agent for discrete path selection, *i.e.,* $m^{k,l}_i \in \{0, 1\}$. Different from those methods, motivated by [30], we relax the discrete space of the gating factor to a continuous one, *i.e.,* $m^{k,l}_i \in [0, 1]$.

Since the gating factor $m^{k,l}_i$ can reflect the estimated probability of the path being enabled, we define the list of weighted output from each path as the router output. This router output is formulated as Eq. 2, where only paths with a positive gating factor are enabled. It is worth noting that, unlike many related methods [23–25, 31, 32], our router allows multiple paths to be enabled simultaneously for a single location.

$$\mathbf{y}^l_i = \{\mathbf{y}^{k,l}_i|\mathbf{y}^{k,l}_i = m^{k,l}_i \cdot f^l_k(\mathbf{x}^l_i), k \in \Omega_i\}, \quad \text{where } \Omega_i = \{k|m^{k,l}_i > 0, k \in \{1, ..., K\}\},\tag{2}$$

As shown in Fig. 3(c), we design a lightweight convolutional network to instantiate the spatial gate $g^l_k(\cdot)$. The input feature is embedded into one channel by using a layer of convolution. Since the

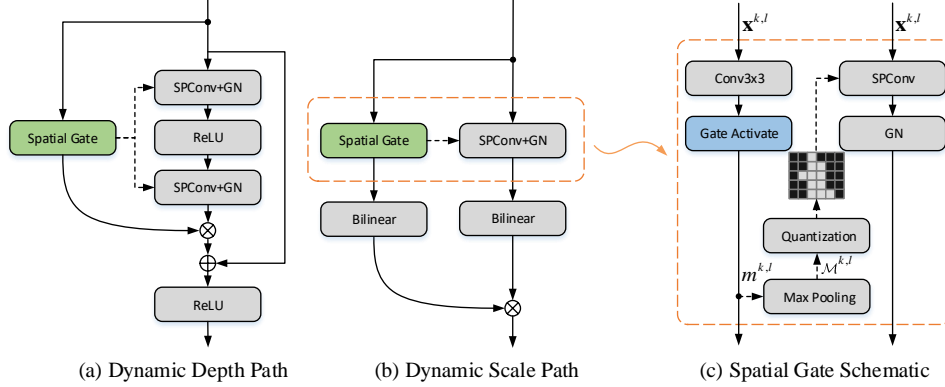

(a) Dynamic Depth Path      (b) Dynamic Scale Path      (c) Spatial Gate Schematic

Figure 3: The diagram of the components in the fine-grained dynamic routing network. 'SPConv' and 'GN' indicate the spatially sparse convolution [27] and the group normalization [35], respectively. Conditional on the input feature, the spatial gate dynamically enables pixel-level locations to be inferred by using spatially sparse convolutions, which is elaborated in the dashed rectangle. The 'Max Pooling', whose stride is one, is used to generate the spatial mask for 'SPConv' according to the size of the receptive field.

output of spatial gate $m_i^{k,l}$ is restricted to the range $[0, 1]$, we propose a new *gate activation* function, denoted as $\delta(\cdot)$, to perform the mapping.

### 2.2.2 Gate Activation Function

To achieve a good trade-off between effectiveness and efficiency, the dynamic router should allocate appropriate computational resources to each location by disabling paths with high complexity. Specifically, for the $i$-th location of the node $l$, the $k$-th path is disabled, when the gating factor $m_i^{k,l}$ output by the gate activation function is zero. Therefore, the range of the gate activation function needs to contain zero. Meanwhile, in order to make the routing process learnable, the gate activation function needs to be differentiable.

For instance, [23, 30, 33] adopts a soft differentiable function (*e.g.,* softmax) for training. In the inference phase, the path whose gating factor below a given threshold is disabled. On the contrary, [34] uses the hard non-differentiable function (*e.g.,* hard sigmoid) in the forward process and the approximate smooth variant in the backward process. However, these strategies cause inconsistency between training and inference, resulting in performance degradation. Recently, a restricted tanh function [26], *i.e.,* $\max(0, \tanh(\cdot))$, is proposed to bridge this gap. When the input is negative, the output of this function is always 0, which allows no additional threshold to be required in the inference phase. When the input is positive, this function turns into a differentiable logistic regression function, so that no approximate variant is needed in the backward process.

However, the restricted tanh has a discontinuous singularity at zero, causing the gradient to change dramatically at that point. To alleviate this problem, we propose a more generic variant of the restricted tanh, which is formulated as Eq. 3 (the visualization is provided in the supplementary material).

$$\delta(v) = \max(0, \frac{\tanh(v - \tau) + \tanh(\tau)}{1 + \tanh(\tau)}) \in [0, 1], \forall v \in \mathbb{R}, \tag{3}$$

Where $\tau$ is a hyperparameter to control the gradient at $0^+$. In particular, Eq. 3 can be degraded to the original restricted tanh when $\tau = 0$. When $\tau$ is set to a positive value, the gradient at $0^+$ will decrease to alleviate the discontinuity problem.

### 2.2.3 Routing Path

As presented in Fig 2, we provide three paths for each fine-grained dynamic router. The same network architecture except for the bilinear operations is adopted in 'Scale (up) Path' and 'Scale (down) Path', which is illustrated in Fig. 3(b). The mode of the bilinear operators is switched to up-sampling and down-sampling for 'Scale (up) Path' and 'Scale (down) Path' respectively, whose scaling factor is set

to 2. The depth of head is critical to the performance, especially for one-stage detectors [2, 3, 20, 21]. To improve the effectiveness of deep network for the head, we use a bottleneck module [28] with a residual connection for the 'depth' path, which is illustrated in Fig. 3(a). To release the efficiency of fine-grained dynamic routing, we adopt the spatially sparse convolutions [27] for each path, which is elaborated in Fig. 3(c). Specifically, the spatial mask $\mathcal{M}^{k,l}$ is generated by performing the max-pooling operation on the gating factors $m^{k,l}$, which is elaborated in Sec. 2.3. The positive values in the spatial mask are further quantified to one. To this end, the quantified spatial mask guides the spatially sparse convolution to only infer the locations with positive mask values, which reduces the computational complexity dramatically. All the spatially sparse convolutions take the kernel size of $3 \times 3$ and the same number of output channels as the input. In particular, to further reduce the computational cost, the convolutions in 'Scale' paths adopt depthwise convolutions [36]. Moreover, we use additional group normalizations [35] to alleviate the adverse effect of gating variations.

## 2.3 Resource Budget

Due to the limited computational resources for empirical usage, we encourage each dynamic router to disable as many paths as possible with a minor performance penalty. To achieve this, we introduce the budget constraint for efficient dynamic routing. Let $\mathcal{C}^{k,l}$ denotes the computational complexity associated with the predefined $k$-th path in the node $l$.

$$\mathcal{B}^l = \frac{1}{N} \sum_i \sum_k \mathcal{C}^{k,l} \mathcal{M}_i^{k,l}, \quad \text{where } \mathcal{M}_i^{k,l} = \max_{j \in \Omega_i^{k,l}} (m_j^{k,l}). \tag{4}$$

The resource budget $\mathcal{B}^l$ of the node $l$ is formulated as Eq. 4, where $\Omega_i^{k,l}$ denotes the receptive field of $i$-th output location in the $k$-th path. As shown in Fig. 3(a), the gating factor $m^{k,l}$ only reflects the enabled output locations in the *last* layer. Since the receptive field of the stacked convolution network tends to increase with depth, more locations in the *front* layer need to be calculated than the last layer. For simplicity, we consider all the locations involved in the receptive field of locations with positive gating factors as enabled, which are calculated by a max-pooling layer, *i.e.*, $\max_{j \in \Omega_i^{k,l}} (m_j^{k,l})$. Furthermore, this function can provide a regularization for the number of connected components. It encourages the connected components of the gating map $m^{k,l}$ to have a larger area and smaller number. This regularization could improve the continuity of memory access and reduces empirical latency [37].

## 2.4 Training Target

In the training phase, we select the appropriate loss functions according to the original configuration of the detection framework. Specifically, we denote the losses for classification and regression as $\mathcal{L}_{cls}$ and $\mathcal{L}_{reg}$, respectively. For the FCOS [3] framework, we further calculate the loss for the centerness of bounding boxes, which is represented as $\mathcal{L}_{center}$. Moreover, we adopt the above-mentioned budget $\mathcal{B}^l$ to constrain the computational cost. The corresponding loss function is shown as Eq. 5, which is normalized by the overall computational complexity of the head.

$$\mathcal{L}_{budget} = \frac{\sum_l \mathcal{B}^l}{\sum_l \mathcal{C}^l} \in [0, 1], \quad \text{where } \mathcal{C}^l = \sum_k \mathcal{C}^{k,l}. \tag{5}$$

To achieve a good tradeoff between efficiency and effectiveness, we introduce a positive hyper-parameter $\lambda$ to control the expected resource consumption. Overall, the network parameters, as well as the fine-grained dynamic routers, can be optimized with a joint loss function $\mathcal{L}$ in a unified framework. The formulation is elaborated in Eq. 6.

$$\mathcal{L} = \mathcal{L}_{cls} + \mathcal{L}_{reg} + \mathcal{L}_{center} + \lambda \mathcal{L}_{budget}. \tag{6}$$

## 3 Experiment

In this section, we first introduce the implementation details of the proposed fine-grained dynamic routing network. Then we conduct extensive ablation studies and visualizations on the COCO dataset [29] for the object detection task to reveal the effect of each component.

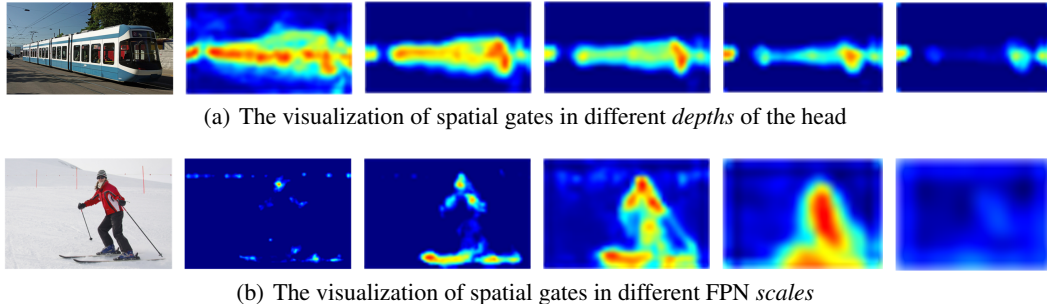

(a) The visualization of spatial gates in different *depths* of the head

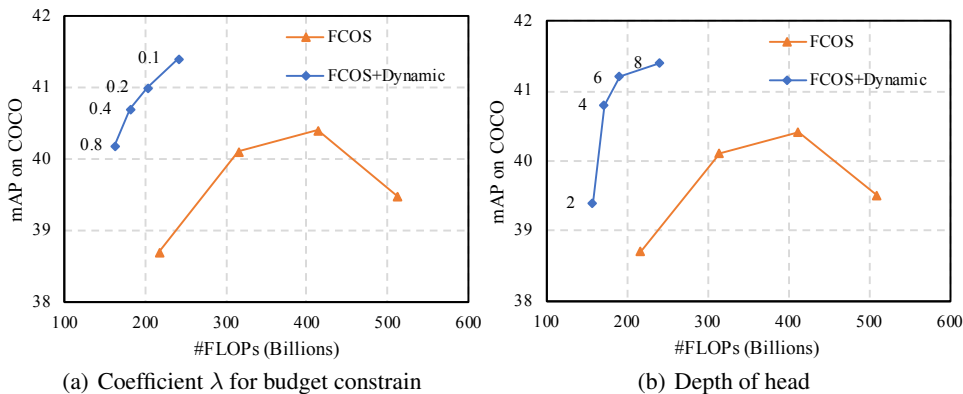

(b) The visualization of spatial gates in different FPN *scales*

Figure 4: Visualization of the spatial gates in dynamic heads. The heatmaps (from left to right) in (a) and (b) correspond to an increase in the depth of a head, and the FPN scale from P3 to P7, respectively. As the depth increase, more and more locations are disabled by the dynamic router to achieve efficiency. Besides, (b) illustrates that the dynamic router can adaptively assign the pixel-level sub-regions of a single instance to different FPN scales.

(a) Coefficient $\lambda$ for budget constrain

(b) Depth of head

Figure 5: The trade-off between efficiency and effectiveness. The FCOS is scaled by the depth of the head (*i.e.,* D2, D4, D6 and D8). Our fine-grained dynamic routing network shows consistent improvements over the baseline FCOS for varying the coefficient of budget constraints (based on FCOS-D8), and the depth of head, respectively.

## 3.1 Implementation Detail

To verify the effectiveness of our proposed method, we conduct experiments on the FCOS [3] framework unless otherwise specified. The FCOS provides a simple architecture that can be used to reveal the properties of our method. In the default configuration of FCOS, it adopts a pair of 4-convolution heads for classification and regression, respectively. Besides, the prediction of centerness shares the same head with the regression branch. For a fair comparison, we replace the original head with a fixed network. It has the same paths as our proposed dynamic head but without dynamic routers. Moreover, $\{\mathrm{D}n|n \in \{2, 4, 6, 8\}\}$ represents the depth of the equipped head.

All the backbones are pre-trained on the ImageNet classification dataset [38]. Batch normalizations [39] in the backbone are frozen. In the training phase, input images are resized so that the shorter side is 800 pixels. All the training hyper-parameters are identical to the 1x schedule in the Detectron2 [40] framework. Specifically, we fix parameters of the first two stages in the backbone and then jointly finetune the rest network. All the experiments are trained on 8 GPUs with 2 images per GPU (effective mini-batch size of 16) for 90K iterations. The learning rate is initially set to 0.01 and then decreased by 10 at the 60K and 80K iterations. All the models are optimized by using Synchronized SGD [41] with a weight decay of 0.0001 and a momentum of 0.9.

Table 1: Comparisons among different settings of the dynamic routers. 'DY' denotes the *dynamic* routing for the path selection, which is coarse-grained by default. 'FG' represents proposed *fine-grained pixel-wise* routing. $\text{FLOPs}_{avg}$, $\text{FLOPs}_{max}$ and $\text{FLOPs}_{min}$ represent the average, maximum and minimum FLOPs of the network. In addition, 'L' and 'H' respectively indicate two configurations with different computational complexity by adjusting the budget constraints.

| Model | DY | FG | mAP(%) | $\text{FLOPs}_{avg}$(G) | $\text{FLOPs}_{max}$(G) | $\text{FLOPs}_{min}$(G) |
|---|---|---|---|---|---|---|
| FCOS [3] | ✗ | ✗ | 38.7 | 216.7 | 216.7 | 216.7 |
| FCOS-D8 | ✗ | ✗ | 39.5 | 512.5 | 512.5 | 512.5 |
| FCOS-D8@L | ✓ | ✗ | 38.6 | 233.0 | 257.6 | 223.5 |
| | ✓ | ✓ | 41.0 | 203.6 | 336.0 | 144.7 |
| FCOS-D8@H | ✓ | ✗ | 41.4 | 471.2 | 481.2 | 452.0 |
| | ✓ | ✓ | **42.0** | 446.6 | 463.1 | 422.7 |

## 3.2 Ablation Study

In this section, we conduct experiments on various design choices for our proposed network. For simplicity, all the reported results here are based on ResNet-50 [28] backbone and evaluated on COCO *val* set. The FLOPs is calculated when given a $1333 \times 800$ input image.

### 3.2.1 Dynamic vs Static

To demonstrate the effectiveness of the dynamic routing strategy, we give the comparison with the fixed architecture in Tab. 1. For fair comparisons, we align the computational complexity with these models by adjusting the coefficient $\lambda$ for the budget constraints in Eq. 6. The results show that the dynamic strategy can not only reduce computational overhead but also improve performance by a large margin. For instance, our method obtains 2.3% absolute gains over the static baseline with lower average computational complexity. As shown in Fig. 4, this phenomenon may be attributed to the adaptive sparse combination of the multi-scale features to process each instance.

### 3.2.2 Fine-Grained vs Coarse-Grained

Different from most of the coarse-grained dynamic networks [23–26, 31], our method performs routing in the pixel level. For comparison, we construct a coarse-grained dynamic network by inserting a global average pooling [42] operator between the 'Conv3 × 3' and the gate activation function in each spatial gate, as shown in Fig. 3 (c). The experiment results are provided in Tab. 1. We find that the upper bound of fine-grained dynamic routing is higher than that of coarse-grained dynamic routing in the same network architecture. Moreover, as the computational budget decreased, the performance of coarse-grained dynamic routing decreases dramatically, which reflects that most of the computational redundancy is in space. Specifically, the fine-grained dynamic head achieves 2.4% mAP absolute gains over the coarse-grained one with only 87% computational complexity.

### 3.2.3 Component Analysis

To reveal the properties of the proposed activation function for spatial gates, we further compare some widely-used activation function for soft routing, which is elaborated in Tab. 2. When adopting the Softmax as the activation function, the routing process is similar to the attention mechanism [43–47]. This means that the hard suppression of the background region is important for the detection task. Meanwhile, as shown in Tab. 3, we verify the effectiveness of the proposed 'Depth' path and 'Scale' path with ablation, respectively. The performance can be further improved by using both paths, which demonstrates that they are complementary and promoting each other.

### 3.2.4 Trade-off between Efficiency and Effectiveness

To achieve a good balance between efficiency and effectiveness, we give a comparison of varying the coefficient $\lambda$ of budget constraints and the depth of equipped head, which is shown in Fig. 5. The baseline FCOS is scaled by the depth of the head. The redundancy in space enables the network to maintain high performance with little computational cost. For instance, when $\lambda$ is set to 0.4, the

Table 2: Comparisons among different activation functions based on the FCOS-D8 framework. Due to the data-dependent property of the dynamic head, we report the average FLOPs here.

| Activation | $\tau$ | mAP(%) | FLOPs(G) |
|---|---|---|---|
| Softmax | - | 40.2 | 512.5 |
| Restricted Tanh | - | 41.4 | 477.6 |
| Proposed | 0.5 | 41.4 | 413.3 |
| | 1.0 | 41.5 | 468.7 |
| | 1.5 | **42.0** | 446.6 |
| | 2.0 | 41.8 | 427.2 |

Table 3: Comparisons of different dynamic routing paths based on the FCOS-D8 framework. 'Scale' and 'Depth' respectively indicate using the proposed dynamic depth path and dynamic scale path for each router. Due to the data-dependent property of the dynamic head, we report the average FLOPs here.

| Scale | Depth | mAP(%) | FLOPs(G) |
|---|---|---|---|
| ✗ | ✗ | 39.5 | 512.5 |
| ✓ | ✗ | 41.5 | 512.0 |
| ✗ | ✓ | 41.3 | 476.0 |
| ✓ | ✓ | **42.0** | 446.6 |

Table 4: Applications on state-of-the-art detectors. 'Dynamic' indicates using the proposed fine-grained dynamic head to replace the original one. For each method, the computational complexity is aligned by adjusting the depth of the head and the budget constrain. All the reported FLOPs are calculated when input a $1333 \times 800$ image, except for 'EfficientDet-D1'. Due to the data-dependent property of the dynamic head, we report the average FLOPs here.

| Method | Backbone | Dynamic | mAP(%) | FLOPs(G) | Params(M) |
|---|---|---|---|---|---|
| RetinaNet [2] | ResNet-50 | ✗ | 35.8 | 249.2 | 34.0 |
| | | ✓ | **38.2** | 245.3 | 47.6 |
| FreeAnchor [21] | ResNet-50 | ✗ | 38.3 | 249.2 | 34.0 |
| | | ✓ | **39.0** | 223.8 | 47.6 |
| FCOS [3] | ResNet-101 | ✗ | 41.0 | 296.0 | 51.4 |
| | | ✓ | **42.8** | 278.9 | 65.0 |
| ATSS [20] | ResNet-101 | ✗ | 41.2 | 296.0 | 51.4 |
| | | ✓ | **43.2** | 289.4 | 65.0 |
| EfficientDet-D1 [5] * | EfficientNet-B1 [48] | ✗ | 38.2 | 6.1 | 6.6 |
| | | ✓ | **38.8** | 5.9 | 9.4 |

\* The FLOPs is calculated when input a $600 \times 600$ image.

proposed network achieves similar performance with the fixed FCOS-D6 network, but only account for about 43% of the computational cost (including the backbone). In particular, without considering the backbone, its computational cost only accounts for about 19% of the FCOS-D6.

## 3.3 Application on State-of-the-art Detector

To further demonstrate the superiority of our proposed method, we replace the head of several state-of-the-art detectors with the proposed fine-grained dynamic head. All the detectors except 'EfficientDet-D1' [5] are trained with 1x schedule (*i.e.,* 90k iterations), and batch normalizations [39] are freezed in the backbone. Following the original configuration, the 'EfficientDet-D1' adopts 282k iterations for training. In addition, all the detectors adopt single-scale training and single-scale testing. The experiment results are presented in Tab. 4. Our method consistently outperforms the baseline with less computational consumption. For instance, when replacing the head of RetinaNet [2] with our fine-grained dynamic head, it obtains 2.4% absolute gains over the baseline.

## 4 Conclusion

In this paper, we propose a fine-grained dynamic head to conditionally select features from different FPN scales for each sub-region of an instance. Moreover, we design a spatial gate with the new activation function to reduce the computational complexity by using spatially sparse convolutions. With the above improvements, the proposed fine-grained dynamic head can better utilize the multi-scale features of FPN with a lower computational cost. Extensive experiments demonstrate the

effectiveness and efficiency of our method on several state-of-the-art detectors. Overall, our method exploits a new dimension for object detection, *i.e., dynamic routing mechanism is utilized for fine-grained object representation with efficiency*. We hope that this dimension can provide insights into future works, and beyond.

## Broader Impact

Object detection is a fundamental task in the computer vision domain, which has already been applied to a wide range of practical applications. For instance, face recognition, robotics and autonomous driving heavily rely on object detection. Our method provides a new dimension for object detection by utilizing the fine-grained dynamic routing mechanism to improve performance and maintain low computational cost. Compared with hand-crafted or searched methods, ours does not need much time for manual design or machine search. Besides, the design philosophy of our fine-grained dynamic head could be further extended to many other computer vision tasks, *e.g.*, segmentation and video analysis.

## Acknowledgments and Disclosure of Funding

This research was supported by National Key R&D Program of China (No. 2017YFA0700800), National Natural Science Foundation of China (No. 61790563 and 61751401) and Beijing Academy of Artificial Intelligence (BAAI).

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
