[Supplementary Material]

# Supplementary Material

## A Visualization

Figure 6: Visualization of the proposed gate activation function. As the coefficient $\tau$ increased, the gradient at $0+$ will decrease to alleviate the discontinuity problem.

Figure 7: Visualization of the spatial gates in a dynamic head. The response maps are generated from three adjacent FPN scales, *i.e.*, P4, P5 and P6. The row and column of the heatmaps correspond to depth and scale, respectively.

(a) Fine-Grained Dynamic Head       (b) Conventional Head       (c) Ground Truth

Figure 8: Comparisons of predictions between the proposed dynamic head and the conventional head. The predictions are generated from the FCOS framework with the specific head when using ResNet-50 backbone.

# B   Runtime

Table 5: The latency and computational complexity of the FPN heads on a Tesla V100 GPU. The computational complexity only accounts for the head.

| Model | Dynamic Head | mAP(%) | Latency$_{avg}$(ms) | FLOPs$_{avg}$(G) |
|---|---|---|---|---|
| FCOS-D6 Baseline | ✗ | 40.4 | 46.8 | 298.1 |
| Ours@Large | ✓ | **41.4** | 53.6 | 117.6 |
| Ours@Small | ✓ | 40.6 | **35.1** | **67.6** |