[Reviews · NeurIPS 2020]

Review 1

Summary and Contributions: This paper proposes a new module, a fine-grained dynamic head, to improve the feature pyramid network (FPN) which is commonly used in many state-of-the-art detectors. At each scale in an FPN, the fine-grained dynamic head conditionally merge pixels from adjacent scales to better detect objects at all scales. It uses a spatial gate to determine which pixels to merge. The spatial gate uses a modified version of activation function from prior work. Experiments show that the fine-grained dynamic head improves the performance of multiple detectors on COCO.

Strengths: This idea of conditionally merging features from different scales is interesting and new in the context of object detection. The authors provide adequate experiments to demonstrate the effectiveness of the fine-grained dynamic head and the contribution of the component in the dynamic head.

Weaknesses: The writing quality of this paper needs to be improved substantially. Section 2 which describes the fine-grained dynamic head, the main contribution of this paper, is confusing and hard to follow. Many of the symbols are not defined clearly in the section which makes it hard to understand the details of the proposed approach. For example, in line 82, what is f^{l}_{k} exactly? What does K represent? Does K represent the number of adjacent feature maps? Furthermore, what is y^{l}_{i} in equation 2? What is y being used? The part about routing path is also confusing. What is dynamic depth path in Fig 3a? Is it the same as the purple line in Fig 2? Is the spatial gate in dynamic depth path to allow the network to just process a subset of the locations in the feature maps? The authors should report the actual inference time instead of FLOPs of their models. Although the proposed approach can improve the performance of state-of-the-art detectors while maintaining similar FLOPs, I do not think the proposed approach actually achieves similar inference speed to the baseline approaches. The authors use sparse convolutions extensively in the fine-grained dynamic head to reduce computational complexity. However, GPUs, the most commonly used devices for inference, do not perform well on sparse data.

Correctness: Yes. But the empirical methodology can be improved to better demonstrate the actual efficiency of the proposed approach.

Clarity: No. The main section of the paper is quite confusing.

Relation to Prior Work: The paper currently does not have a related work section. It discusses the differences briefly in the approach section. The differences are clear to me. But I would recommend the authors to have a dedicated section to discuss the difference in detail.

Reproducibility: No

Additional Feedback:


Review 2

Summary and Contributions: This paper proposes a fine-grained dynamic head which selects features from different feature scales for different sub-regions. Experiments on COCO show that the proposed method outperforms different single-stage object detector baselines.

Strengths: 1. The proposed method is interesting. 2. Promising results are obtained by the proposed method.

Weaknesses: 1. Important reference. The motivation of this paper is quite similar to the motivation in the previous work [a]. In [a], the authors also propose a method to aggregate features from different feature scales for different sub-regions. The authors should discuss and compare with [a] in detail. 2. Experiments. 1) Since this paper and the paper [a] share similar motivations, it would be better to show result comparisons between the methods from this paper and [a]. 2) For a fair comparison, it would be better to show results of FCOS by using SPConv+GN as the head. 3) Since FCOS-D6 obtains better results than FCOS-D8 (from Figure 5), it would be better to compare FCOS-D8@H/FCOS-D8@L with FCOS-D6 in Table 1 to show the real improvements from the proposed method. [a] PointRend: Image Segmentation as Rendering

Correctness: Yes

Clarity: Most parts of this paper are well written. It would be better to give explanations of y_{i}^{l} in Equation (2) in main paper texts instead of only showing in Figure 2.

Relation to Prior Work: It would be better to discuss with the previous work [a]. See Weaknesses for more details. [a] PointRend: Image Segmentation as Rendering

Reproducibility: Yes

Additional Feedback: The authors have addressed my concerns in their rebuttal. So I would keep my accept rating to this work.


Review 3

Summary and Contributions: This paper proposed a fine-grained dynamic head to conditionally select a pixel-level combination of FPN features from different scales for each instance. This method utilized the fine-grained dynamic routing mechanism to improve detection performance and keep the low computational cost. The experimental results demonstrate the effectiveness of the proposed method.

Strengths: ++ The cleansing method is technically sound. The design of dynamic mechanism increases the ability of learning multi-scale representations and improves object detection performance. ++ The experimental results are promising. There are quite a few ablation studies. ++ This paper is clearly presented.

Weaknesses: -- The novelty is relatively restricted. Applying dynamic routing to networks is already explored for semantic segmantation [20]. To me, this work seems like an extension of [20] to pixel-level routing, since the overall framework is very similar. Also, the adopted spatially sparse convolution is not proposed the this paper. Hence, the major technical novelty seems to be the pixel-level extension (which is not difficult to think of and can be realized quite straightforwardly) and the activation function. -- I am not sure if the performance of FCOS baselines in Figure 5 is consistant with the original FCOS paper. For example, the original FCOS achieves 37.1 (and 38.6 for the improved version). The implementation details of FCOS-D{2,4,6} should be clearly described. -- There is an improved version of FCOS (with 4 tricks added). It is better to reported results based on the improved FCOS as the absolute AP improvement of the proposed method may not be the same between the two versions. -- The improvement of the proposed activation function is small (0.4 AP) compared to Restricted Tanh. -- I would like to see the latency of the proposed method comparing to those baseline counterparts, since the FLOPS are not always equivalent to the latency. Especially, I suspect that the proposed method is not latency-friendly, because the computation of each FPN level is no longer independent. For example, the computation of P5 depends on P4, which would be slower than P5 alone.

Correctness: The claims and methods are correct.

Clarity: The paper is in general well written.

Relation to Prior Work: The difference between the paper and previous works is clearly discussed.

Reproducibility: Yes

Additional Feedback:


Review 4

Summary and Contributions: The paper proposes a novel fine-grained dynamic routing mechanism for object detection where it conditionally selects features from multiple FPN scales for each pixel-level sub-region of an instance by using the data-dependent spatial gates. It dynamically allocates pixel-level sub-regions to different resolution stages and aggregates them for the finer pixel-level feature representation. To reduce computational complexity, spatially sparse convolution is used in combination with a newly proposed activation function based on restricted tanh. Depth path uses bottleneck module with residual connection. Scale paths use common topology except for the bilinear operation. Output of the spatial gate is the gating factor, which is continuous, representing an estimate of probability of path being enabled. A more generic variant of the restricted tanh for the gate activation function is proposed for better gradient behavior at 0. Spatially sparse convolution is used for efficiency, based on max pooling of gating factors. A budget constraint is used in the loss function to encourage each dynamic router to disable as many paths as possible with a minor performance penalty Dynamic strategy achieves consistent improvements with little computational overhead (2.6% mAP absolute gains on COCO with much lower computational complexity). Fine-grained dynamic routing give 0.8% mAP absolute gain over dynamic coarse grained (using global average pooling) with 71% computational complexity. After Rebuttal: ---------------- Concern about FCOS baselines was the one that stood out for me after reading R3's review, and the authors recognize this and mention that they will address this. Same rating as before.

Strengths: The paper proposes a novel fine-grained pixel-level dynamic routing head as well a new activation function. Computation complexity was reduced by using gating factors in combination with spatially sparse convolution as well using a computational budget term in the loss function. Ablation study is extensive with good insights. (1) static vs dynamic coarse vs dynamic fine routing (2) new activation function (3) scale, depth dynamic routing paths (4) 5 SOTA detectors Ablation study highlights the effectiveness of fine-grained dynamic routing to improve mAP and also lowering the total computational complexity. It also shows that it works across various key SOTA approaches.

Weaknesses: The value of the new activation function as in Table 2 is weak. So this is not a big contribution. Other parts seem fine.

Correctness: Yes.

Clarity: Paper is well written with extensive ablation study and insights.

Relation to Prior Work: Yes, they appear in the context of each section.

Reproducibility: Yes

Additional Feedback: Related works are not well grouped in a single section. Some relevant works appear in the context which is helpful. Minor typo: Line 69 (advantage)

[Author Response · NeurIPS 2020]

# Author Rebuttal for NeurIPS 2020 Submission #3770

We thank all the reviewers for their valuable comments and suggestions. To improve readability, we will add a section of the related work. Besides, we will carefully revise the manuscript and describe the idea more clear according to the suggestions. We respond to the main concerns as follows. All the source code will be released to the community soon.

**Q:** *The authors should report the actual inference time/latency of their models.*

**A:** We evaluate the proposed method on a Tesla V100 GPU. Since our method only modifies the FPN head, we report the latency and the computational complexity of that part in Tab. 5. The result demonstrates the efficiency of our method on GPUs even with unfriendly sparse operations. Besides, our method has great potential to further improve efficiency by leveraging a sophisticated optimization or using specialized hardware accelerators for sparse operations (*e.g.*, SCNN, Cambricon-X, EIE and Eyeriss V2). We will clarify it and report the empirical runtime in the final version.

Table 5: The latency and computational complexity of the FPN heads on a Tesla V100 GPU. 'DH' is the dynamic head.

| Model | DH | mAP(%) | $\text{Latency}_{avg}$(ms) | $\text{Latency}_{max}$(ms) | $\text{Latency}_{min}$(ms) | $\text{FLOPs}_{avg}$(G) |
|---|---|---|---|---|---|---|
| FCOS-D6 Baseline | ✗ | 40.3 | 46.8 | - | - | 298.1 |
| Ours@Large | ✓ | **41.0** | 52.3 | 63.6 | 42.9 | 104.3 |
| Ours@Small | ✓ | 40.3 | **34.9** | 46.5 | 28.9 | **70.0** |

## Response to Reviewer #1

**Q1:** *Some confusions on statements of symbols.*

**A1:** Sorry for the confusion. Specifically, $K$, $f^{l,k}$, and $y_i^l$ represent the number of adjacent FPN scales, an adjacent feature map of node $l$ and the feature vector of a pixel in the output of node $l$, respectively. We promise to revise the manuscript according to your suggestions carefully.

**Q2:** *What is the dynamic depth path in Fig.3a? Is it the same as the purple line in Fig.2?*

**A2:** Yes. The dynamic depth path in Fig.3a is the same as the purple line in Fig.2, whose spatial gate allows the network to just process a subset of locations. We will further explain this point in the final version.

## Response to Reviewer #2

**Q1:** *The authors should discuss and compare with the "PointRend".*

**A1:** Thanks for your suggestion. Although the motivation has some similarities, "PointRend" mainly focuses on the boundary refinement in the instance segmentation task without high-level semantic enhancement. The reported performance (refer to the repo in GitHub) on object detection is also inferior to ours. We will add more discussions.

**Q2:** *It would be better to show results of FCOS by using SPConv+GN as the head.*

**A2:** The SPConv is identical to the regular convolution when enabling all the locations, and the GN is adopted by default. Please refer to Tab.1, Tab.2 and Sec.3.1 for more details. We will clarify it in the final version.

## Response to Reviewer #3

**Q1:** *Some concerns about the novelty.*

**A1:** Sorry for the misunderstanding. Different from the dense prediction in the semantic segmentation task, the prediction of object detection is encouraged to be sparse in space, *e.g.*, the number of foreground samples (1.17% for FCOS on COCO dataset) is much less than that of background. Therefore, a large amount of computation in space is redundant and can be eliminated by the pixel-level dynamic routing. Meanwhile, pixel-level aggregation can handle the small but representative sub-regions of an instance, which is crucial to high-quality object detection. However, *these properties could not be achieved by the feature-level routing methods, e.g., [20].* This paper is the first to introduce the pixel-level dynamic routing mechanism into object detection, which brings stronger multi-scale representation with less computational complexity. Besides, the pixel-level extension is not straightforward. It needs to consider some extra aspects, *e.g.*, the change of the receptive field and the effect of regional connectivity on efficiency (refer to Sec.2.3).

**Q2:** *Some concerns about the FCOS baselines.*

**A2:** Our implementation for FCOS does not use the centerness-weighted trick for the regression loss, which can obtain 0.5 mAP absolute gains. We will update the results with this trick in the final version. The baseline head of FCOS-D{2,4,6,8} adopts the same architecture as dynamic head with depth and scale paths, but without spatial gates.

**Q3:** *The improvement of the proposed activation function is small compared to Restricted Tanh.*

**A3:** Please refer to "A1" in the section "Response to Reviewer #4".

## Response to Reviewer #4

**Q1:** *The value of the new activation function as in Table 2 is weak.*

**A1:** Thanks for this comment. Our major contribution is to introduce the pixel-level dynamic routing mechanism into object detection, which enhances sub-region features of an instance with less computational complexity. The new activation function can provide a better and more generic solution to realize end-to-end training for spatial gates.

[Meta-Review · NeurIPS 2020]

The idea of conditionally merging features from different scales in FPN is interesting as acknowledged by all reviewers, and promising results are obtained by the proposed method. The AC agrees with the reviewers that the paper passes the acceptance bar of NeurIPS and recommends acceptance. The authors should add the clarifications and discussion in the rebuttal to the camera-ready version.